# Reliability of Binocular Esterman Visual Field Test in Patients with Glaucoma and Other Ocular Conditions

**DOI:** 10.3390/diagnostics14040433

**Published:** 2024-02-16

**Authors:** Shuhei Fujimoto, Kengo Ikesugi, Takako Ichio, Kohei Tanaka, Kumiko Kato, Mineo Kondo

**Affiliations:** Department of Ophthalmology, Mie University Graduate School of Medicine, Tsu 514-8507, Japan; pooheiei25@gmail.com (S.F.);

**Keywords:** binocular Esterman visual field test, test–retest reliability, visual disability, peripheral vision

## Abstract

The binocular Esterman visual field test (EVFT) of 120 points was the first method to quantify the defects in the binocular visual field. It is used in many parts of the world as a standard test to determine whether an individual has the visual capabilities to drive safely. In Japan, it is required for the grading and issuance of visual disability certificates. The purpose of this study was to determine the reliability of the EVFT results. We studied 104 patients who had undergone the binocular EVFT at Mie University Hospital. Their mean age was 68.0 ± 11.4 years, and the best-corrected visual acuity of the better eye was 0.18 ± 0.38 logMAR units. The EVFT was performed twice on the same day, and the results of the first and second tests were compared. The mean Esterman scores for the first and second test were 89.3 ± 30.5 and 89.1 ± 30.2, respectively, and the test times were 338.9 ± 86.8 and 336.7 ± 76.4 s, respectively. The differences were not significant (*p* = 0.69 and *p* = 0.33). In the Bland–Altman analyses (second–first test) of the Esterman scores, the mean difference was 0.38 without significant fixed errors (*p* = 0.20) or proportional errors (*p* = 0.27). The limits of agreement within the 1.96 standard deviation were −8.96 to +9.45 points. The agreement rate for the most peripheral 24 test points was significantly lower than the agreement rate for the other 96 test points (*p* < 0.01). The agreement rate of the upper visual field was significantly lower than that of the lower field (*p* < 0.01). The overall reliability rate of the EVFT is acceptable, but the peripheral and upper test points have relatively low reliability rates. These findings are important for interpretations of the EVFT results.

## 1. Introduction

The binocular Esterman visual field test (EVFT), first introduced by Benjamin Esterman in 1982 [1], has become a standard test for determining binocular visual field deficits [2,3,4,5,6]. Esterman’s initial grid was introduced in 1967, and it was based on a monocular tangent screen examination which quantified the visual field defects by the number of points included [7].

In 1968, Esterman developed a second monocular grid based on the monocular Goldman perimeter [8]. This version had 100 points with a radius of 50° nasally and 80° temporally, with no points within the central 5° radius. The final version of the scoring system, presented in 1982, merged the two monocular grids into one binocular grid that consisted of 120 test points. This allowed for a more comprehensive assessment of the binocular visual field [1].

It is essential to know that relying solely on the EVFT for critical determinations can be limiting. The characteristics of the disease being tested, the stage of the disease, and the specific visual field abnormalities necessitate the use of a variety of automated visual field tests. Despite these considerations, the EVFT still plays a significant role in diagnosing and managing ocular disorders in ophthalmology and optometry. It remains one of the critical tests for evaluating the integrity of the visual field, especially the peripheral visual field. The results of this test are needed to diagnose and monitor ocular disorders such as glaucoma and retinitis pigmentosa [9,10]. Its simplicity and compatibility with the results of other perimeters have led to its wide-spread use. The test has been used to assess automobile driving capabilities [2,11], and it is a requisite for issuing visual disability certificates in some countries. The application of the EVFT varies globally, and it is influenced by different healthcare systems, cultural backgrounds, and legal frameworks. This diversity results in different interpretations and uses of the test results, especially in the context of visual disability certification and access to public services. For instance, in Japan, the interpretation of the Esterman score for visual impairment classification is specific to its healthcare system. An Esterman score of 100 or less is categorized as a grade 5 visual impairment, while a score of ≤70, which indicates more severe visual field impairment, is classified as grade 2 to 4 [12].

Despite the long history and widespread use of EVFTs, its test–retest reliability has not been extensively studied. This is important because inconsistent results could lead to misdiagnosis or underestimations or overestimations of the visual impairments of an individual.

This absence of research is particularly significant, as reliable results are the foundation of a correct diagnosis and effective evaluations of visual impairments.

Therefore, a careful exploration of the test–retest reliability of the EVFT is a necessary step towards ensuring its continued trustworthiness in clinical and public health settings. Given the limited reports on the reliability of EVFTs, the purpose of this study was to determine the reliability of EVFTs.

## 2. Materials and Methods

We studied the medical records of 104 patients who had undergone the binocular EVFT at Mie University Hospital between January 2022 and December 2022; 98 of the patients had been diagnosed with glaucoma, and 6 were diagnosed with other retinal or optic nerve diseases. Because this was a retrospective study, the majority of cases undergoing the EVFT in our routine practice were glaucoma cases. The mean age of our patients was 68.0 ± 11.4 years, and there were 46 women and 58 men. The mean ± SD of the best-corrected visual acuity (BCVA) of the better eye was 0.18 ± 0.38 logMAR units.

The EVFT was performed using the Humphrey Field Analyzer 3 (HFA3, Carl Zeiss Meditec, Dublin, CA, USA). The EVFT examined more than 130° visual field and consisted of 120 test points in a suprathreshold manner with a size III white stimulus at an intensity of 10 dB. False positive and false negative responses were accessed in a similar fashion to the monocular programs. In the binocular mode, the video eye monitor was aligned to the bridge of the nose; thus, the stability of the fixation was monitored indirectly by observation. However, all participants had undergone previous visual field examinations so that participants with a history of poor fixation were excluded from the study. The subjects were instructed to focus on a static fixation target at the center while keeping their head in a primary position. They were asked to press a response button as soon as they noticed a light spot appearing on the screen in front of them. Before starting the test, subjects were given the opportunity to practice. The EVFT was conducted twice in all cases, and the results of the first and second tests were compared. In all cases, the two tests were conducted by the same examiner within a team of five certified orthoptists, each with over one year of experience. The EVFT tests were generally conducted in two consecutive sessions. However, appropriate break intervals were arranged at the request of the patient to accommodate their comforts and needs.

To examine the reliability of each test point, the 120 test points in the EVFT were numbered in an order that considered the four quadrants around the center point. The upper quadrants were considered positive on the Y-axis, while the lower quadrants were considered negative. Similarly, the right quadrants were positive on the X-axis, and the left quadrants were negative. Numbering was performed by the values on the Y-axis decreasing, or by the ascending X-axis values if the Y-axis values were the same (Figure 1). The coordinates for each of the 120 EVFT locations, provided by Carl Zeiss Meditec, are shown in Table 1.

Based on the findings of prior studies, participants with false positive or negative rates exceeding 33% were excluded from the study [13,14]. We focused on a comprehensive analysis of various aspects of EVFTs. This included a detailed examination of the mean Esterman score and the duration to perform the first and second EVFT. In addition, we used Bland–Altman analyses to assess the agreement of the Esterman scores. Multivariate analyses were used to identify the factors influencing the reliabilities of the EVFTs. Additionally, the concordance of the results across each of the 120 test points within the EVFT was carefully evaluated.

Statistical analysis was performed using IBM SPSS for Windows Ver 28.0.1.0 (IBM Corp., Armonk, NY, USA). The Shapiro–Wilk test was performed to validate the assumption of normal distribution. Paired *t*-tests were used to determine the significance of the differences in the first and second Esterman scores and test times. Agreement between two sets of EVFT scores was evaluated through Bland–Altman analysis. Multiple regression analysis examined factors influencing the differences in the scores of the two tests considering three independent variables. The Mann–Whitney U-test was used to evaluate the reliability rate across the test points of the EVFT. For all tests, a *p* < 0.05 was considered statistically significant.

## 3. Results

The results of the first and second Esterman scores and testing times for the 104 subjects are shown in Table 2. The numbers are the means ± standard deviations. There were no significant differences between the first and second Esterman scores and test times (*p* = 0.69, *p* = 0.33, respectively; paired *t*-tests).

A Bland–Altman plot demonstrating agreement between the two sets of Esterman visual field test scores is presented in Figure 2. The solid line at zero is the point of no differences. The plot reveals a mean difference represented by the dashed line at 0.38. This indicated nonsignificant systematic discrepancies between the two tests. The limits of agreement, which span from −8.96 to +9.45 points, are marked by the outer dashed lines. There is no significant fixed error (*p* = 0.20) or proportional error (*p* = 0.27), indicating that the measurements are reliable across the range of scores.

Table 3 presents the results of a multiple regression analysis that examined the factors influencing the consistency between the first and second Esterman scores. The dependent variable is the difference in EVFT scores between the two tests, and the analyses considered age, sex, and visual acuity as independent variables. The results indicated that age with a standardized regression coefficient of −0.142 and a *p* = 0.137, and sex (0 for males and 1 for females) with a coefficient of 0.037 and a *p* = 0.156, had nonsignificant impacts on the test consistency. The BCVA (logMAR units of the better eye) also had a nonsignificant effect, with a standardized regression coefficient of 0.066 and a *p* = 0.510. The coefficient of determination (R^2^) was 0.024, which suggests that these variables collectively explain a very small proportion of the variance in the difference between the two Esterman test scores.

Figure 3 is a visual representation of the reliability rate (%) across the test points in the Esterman visual field test. The color map illustrates variations in the agreement rates. Deep blue indicates higher rates of concordance, while deep red indicates lower rates of concordance with a mean consistency rate of 89.8 ± 4.6%. Additionally, the figure includes a table on the right which displays the number of test points corresponding to each color-coded concordance rate. The most consistent range was 91–92% in 27 points. Point number 55 had the highest agreement rate of 98.1%, while point number 44 had the lowest rate of 74.0%. The most peripheral 24 test points had a significantly lower consistency rate of 84.3 ± 3.4% in contrast to the central 96 points which had a rate of 91.1 ± 3.8% (*p* < 0.01, Mann–Whitney U-test). The agreement rates also differed significantly between the upper (88.0 ± 3.8%) and lower visual fields (90.6 ± 4.8%; *p* < 0.01, Mann–Whitney U-test), indicating a trend of greater reliability in the lower visual field.

## 4. Discussion

We examined the reliability of two EVFTs, an automated visual field analyzer. The EVFT is a mandatory driving assessment test in many countries [10,11,15]. It has gained additional significance as it has become a critical test for the certification of visual impairment and is drawing increased attention in the clinic as it has been used in Japan since 2018 [12]. Despite its increasing use in automated visual field examination, its reliability has not been extensively reported.

The EVFT has become a standard test for assessing binocular visual field deficits and was a significant advancement in the field of ophthalmology. This test is highly regarded and is widely accepted in many countries for determining an individual’s visual capability to drive, especially in cases where peripheral vision might be compromised due to medical conditions, such as in advanced glaucoma, diabetes, stroke, head injuries, and brain tumors [11]. This approach allows for the assessment of the binocular visual field to be performed in a comprehensive manner. The significance of this test lies in its ability to offer a more realistic measure of functional vision compared to monocular testing. It considers natural binocular enhancement, where two seeing eyes compensate for defects in their fellow eyes, thus providing a more accurate representation of the visual capabilities of an individual in real-world situations [11,16].

While the EVFT is recognized for its versatility, its reliability in clinical settings has not been extensively examined, probably due to its infrequent application in disease management. This gap in determining its reliability was the motivation for this study.

The results showed that the Esterman scores and test times were not significantly different between the results of the first and second tests. Research on the measurement times of the EVFT is limited, but there is some relevant information available. In a study of patients with choroideremia and Stargardt disease in age- and sex-matched controls, it was found that each standard Esterman test took approximately 3–6 min to complete [16]. The report did not include any mention of the reliability of the tests.

Our application of Bland–Altman analysis of consecutive EVFTs found a mean difference of 0.38, with the limits of agreement ranging between −8.96 and +9.45 points. These results indicated a lack of significant fixed or proportional errors, suggesting that factors such as training effects or fatigue did not influence the results of the two consecutive EVTF outcomes. It is important to discuss the practical significance of these variations, specifically whether the thresholds we have selected would change the assessment of the visual function in real-world situations. For example, while our analyses did not show a significant correlation between the EVFT score variability and the visual field scores, the narrow margin of error near the critical thresholds of 100 and 70 points could have significant implications. Minor variations in the EVFT results might lead to changes in the grading of impairments and affect the patient’s eligibility for public welfare services or driving licenses. This underlies the necessity of considering how EVFT score variations are interpreted, especially where these classifications have considerable effects on an individual’s lifestyle and access to services. Our findings highlight the importance of accurate and reliable EVFT evaluations, not only in clinical practice but also in policy formulations on visual impairment assessments.

Multivariate analyses were performed for age, sex, and visual acuity as factors that may have contributed to the variability of EVFT scores. However, these factors were found to not have a significant effect on EVFT scores. Thus, the results did not indicate that older individuals or those with poorer visual acuity had more variations in the EVFT results. These observations prompted us to consider that there might be other factors that could affect the reliability of the EVFT, possibly including cognitive aspects or the varying stages of different diseases. On the other hand, these findings support the conclusions drawn from the Bland–Altman analysis. Despite the low Esterman scores indicating visual impairments, our results suggested that such impairments did not significantly affect the reliability of the scores of the affected individuals. On the other hand, Birt and colleagues studied 106 glaucoma patients who underwent Humphrey automated visual field testing [17]. They reported that the severity of glaucomatous visual field defects and age were factors influencing the reliability of the Humphrey visual fields. Similarly, Kartz reported that the short-term and long-term variations were greater in the eyes of older individuals [18]. Most of our participants had prior experience with automated perimetry, including the 30-2 threshold program, and the average age of our subjects was 68.0 ± 11.4 years, a predominantly elderly cohort. These factors were considered in the reliability scores, and we found no significant correlations between age or visual acuity of the better eye and the reliability of EVFT scores. These observations are significant despite the lack of monitoring of the fixation during the EVFT and the test methods, which involved keeping both eyes open. These factors may have influenced the results.

We numbered the EVFT measured point from 1 to 120. The average agreement rate of individual test points was 89.8%, and the highest agreement rate was found at point 55, and the lowest agreement rate was found at point 44. Interestingly, the test point with the lowest agreement rate corresponded to Marriott’s blind spot of the left eye. In addition, the measurement number with the lowest five agreement rates was number 44 followed by 119 (76.0%), 2 (78.8%), 47 (79.8%), and 1 (79.8%), with number 47 being the test point corresponding to Marriott’s blind spot of the right eye. Pasino and colleagues discussed the relationship between binocular vision and specific eye conditions but did not consider Marriott’s blind spot [19]. Reche-Sainz et al. also reported on binocular vision in glaucomatous eyes and examined binocular visual function impairments in patients with glaucoma. They identified several key factors. For example, the suppression of central vision in distant vision, and a loss of stereoacuity. However, the study did not address the potential involvement of Mariotte’s blind spot in binocular vision in these visual impairments [20].

The agreement rates for the most peripheral 24 test points were significantly lower than the agreement rates for the other 96 test points. The agreement rates of the upper visual field were significantly lower than the agreement rates of the test points in the lower field. The reason for the high proportion of points with low agreements at the edges of the left and right visual fields may also be because they are outside the visual field range of the contralateral eye. The normal extent of the peripheral field of vision from the center is about 50 degrees superiorly, 70 degrees inferiorly, 60 degrees nasally, and 100 degrees temporally [21]. It is also possible that the lower test reliability in the peripheral visual field segment was due to a lower sensitivity of the peripheral visual field compared to the central field. The points with the highest concordance rates were relatively more frequent in the inferior part of the central region. Most of the subjects were patients with glaucoma, a condition characterized by upper visual field defects [22,23,24]. It is possible that the relatively fewer points of visual field defects in the inferior part contributed to the better reliabilities. These findings suggested that for the EVFT, considering the exclusion of the most peripheral test points or omitting them from the assessment scores might be a viable strategy. Such modifications could potentially increase the test’s overall reliability and provide more clinically pertinent data. However, the implications of these changes on diagnostic accuracy necessitate further investigations, highlighting the need for a balanced approach in integrating these findings into clinical practice.

There are a few limitations to this study, especially its retrospective nature. The predominance of glaucoma among the diseases studied may have biased the interpretation of the results. Additionally, the fact that almost all subjects had prior experience with automated visual field testing could have influenced the outcomes.

Despite these limitations, our study contributes important findings, informing on the test–retest reliability and agreement of EVFT scores, including an extensive analysis of each of the 120 test points. We acknowledge that the limitations of our study might affect the generalizability of our findings. Future research should address these limitations to enhance the applicability of EVFT in different clinical settings. Specifically, expanding the range of diseases studied beyond glaucoma, collecting more detailed data on patient backgrounds, and analyzing factors that influence reliability are critical areas of focus for future studies. Such research endeavors would not only address the identified limitations of our current study but also contribute significantly to the broader application and understanding of EVFT in various clinical scenarios.

## 5. Conclusions

In conclusion, our study reveals that the overall reliability rates of the EVFT are generally satisfactory, and age, sex, and visual acuity have minimal impact on the reliability of the test. Significant variability exists particularly in peripheral and upper test points. This variability is a crucial consideration for accurate interpretation in clinical settings, as the average agreement rate drops from 89.8% in individual test points in these specific areas. Such variability in reliability could influence patient management and diagnostic decision making, such as visual impairment classifications specific to public healthcare systems. Recognizing and addressing these disparities in the reliability rates and partial discrepancies will not only enhance EVFT’s clinical application but also guide future research. There is a potential for revising the appropriate placement of test points and for reevaluating the Esterman score’s weighting distribution and the overall assessment program. These revisions could lead to the development of a more nuanced and effective approach to visual field testing, ultimately resulting in more accurate and dependable assessments of visual fields.

## Figures and Tables

**Figure 1 diagnostics-14-00433-f001:**
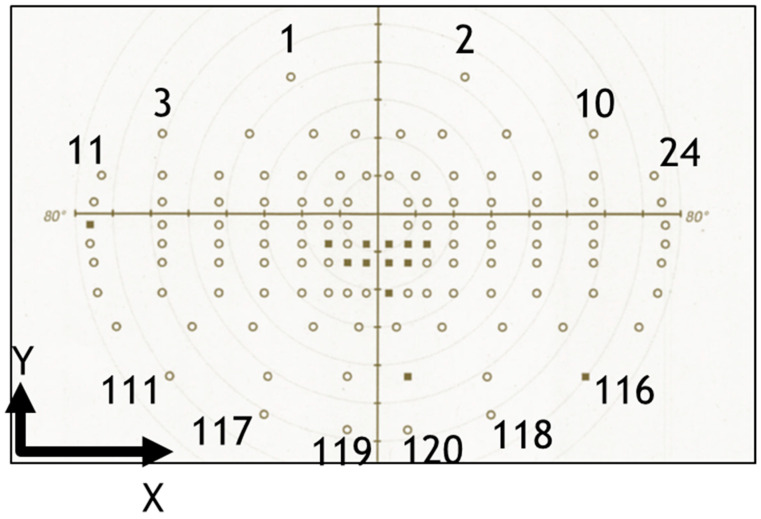
The binocular Esterman visual field test (EVFT). The 120 EVFT test points are numbered in the order of descending y-axis values, and ascending x-axis value if the y-axis values were the same. The white circles represent points that are seen, and the black squares represent points that are missed.

**Figure 2 diagnostics-14-00433-f002:**
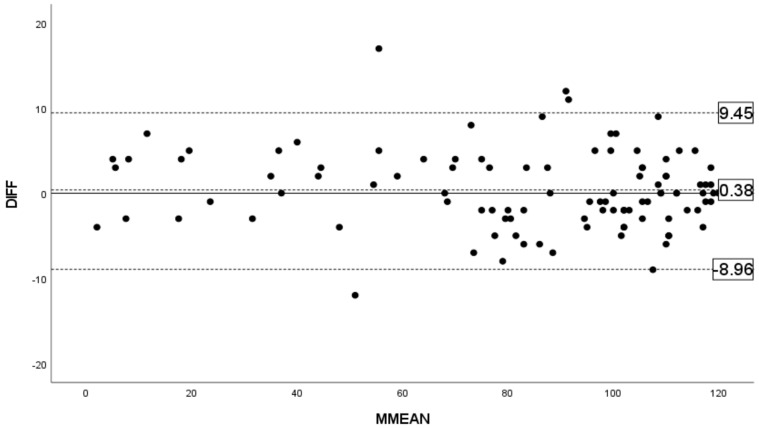
Bland–Altman analysis of the Esterman scores. Each black dot represents the test result of an individual case. The x−axis, MMEAN, represents the mean of measurements from the first and second EVFT. The y−axis, DIFF, represents the difference in scores between the first and second tests for each individual pair of measurements (2nd−1st). The mean difference was 0.38 without a significant fixed error or proportional error. The limits of agreement were −8.96 to +9.45 points.

**Figure 3 diagnostics-14-00433-f003:**
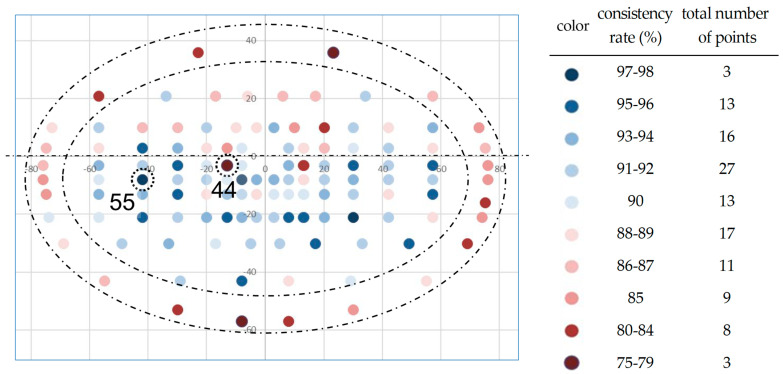
Color map of EVFT consistency rate by test point. The highest agreement rate was found in point number 55 (98.1%), and the lowest agreement rate was found in point number 44 (74.0%). The agreement rate for the most peripheral 24 test points (area enclosed by the dashed circle line) was significantly lower than that of agreement rate for the other 96 test points. The agreement rates of the upper visual field (38 test points) were significantly lower than the agreement rates of the lower field (82 test points), which is divided by dashed line.

**Table 1 diagnostics-14-00433-t001:** The coordinate points of the binocular Esterman visual field test (EVFT). The table numbers are the X−and Y−axis coordinates for each of the 120 test points. The unit is in degrees. The x-axis values represent the horizontal coordinates, while the y-axis values represent the vertical coordinates. Each test point number corresponds to a specific location in the visual field.

EVFT test point	1	2	3	4	5	6	7	8	9	10	11	12	13	14	15	16	17	18	19	20
X	−23	23	−57	−34	−17	−6	6	17	34	57	−73	−57	−42	−30	−20	−10	−3	3	10	20
Y	36	36	21	21	21	21	21	21	21	21	10	10	10	10	10	10	10	10	10	10
EVFT test point	21	22	23	24	25	26	27	28	29	30	31	32	33	34	35	36	37	38	39	40
X	30	42	57	73	−75	−57	−42	−30	−20	−13	−8	8	13	20	30	42	57	75	−76	−57
Y	10	10	10	10	3	3	3	3	3	3	3	3	3	3	3	3	3	3	−3	−3
EVFT test point	41	42	43	44	45	46	47	48	49	50	51	52	53	54	55	56	57	58	59	60
X	−42	−30	−20	−13	−8	8	13	20	30	42	57	76	−76	−57	−42	−30	−20	−13	−8	−3
Y	−3	−3	−3	−3	−3	−3	−3	−3	−3	−3	−3	−3	−8	−8	−8	−8	−8	−8	−8	−8
EVFT test point	61	62	63	64	65	66	67	68	69	70	71	72	73	74	75	76	77	78	79	80
X	3	8	13	20	30	42	57	76	−75	−57	−42	−30	−20	−13	−8	−3	3	8	13	20
Y	−8	−8	−8	−8	−8	−8	−8	−8	−13	−13	−13	−13	−13	−13	−13	−13	−13	−13	−13	−13
EVFT test point	81	82	83	84	85	86	87	88	89	90	91	92	93	94	95	96	97	98	99	100
X	30	42	57	75	−74	−57	−42	−30	−20	−13	−8	−3	3	8	13	20	30	42	57	74
Y	−13	−13	−13	−13	−21	−21	−21	−21	−21	−21	−21	−21	−21	−21	−21	−21	−21	−21	−21	−21
EVFT test point	101	102	103	104	105	106	107	108	109	110	111	112	113	114	115	116	117	118	119	120
X	−69	−49	−33	−17	−5	5	17	33	49	69	−55	−29	−8	8	29	55	−30	30	−8	8
Y	−30	−30	−30	−30	−30	−30	−30	−30	−30	−30	−43	−43	−43	−43	−43	−43	−53	−53	−57	−57

**Table 2 diagnostics-14-00433-t002:** Esterman scores and test times. The numbers are the means ± standard deviations. The first and second Esterman scores and test times were not significantly different, with *p* = 0.69 and 0.33, respectively; paired *t*-tests.

	Esterman Score	Test Time (s)
1st test	89.3 ± 30.5	338.9 ± 86.8
2nd test	89.1 ± 30.2	336.7 ± 76.4

**Table 3 diagnostics-14-00433-t003:** Multiple regression analysis of the differences between the 1st and 2nd Esterman score. The age, sex, and visual acuity of the better eyes had little influence on the reliability of the Esterman test scores. (R^2^ = 0.024).

	Regression Coefficient	Standardized Regression Coefficient	Significance (*p*-Value)
Age	−0.069	−0.142	0.137
Sex (Male: 0 Female: 1)	0.416	0.037	0.156
Visual acuity (LogMAR, better eye)	0.974	0.066	0.510

## Data Availability

The data that support the findings of this study are available from the corresponding author (K.I.) upon reasonable request, due to restrictions of privacy.

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
