# Peer review of "Reliability of Binocular Esterman Visual Field Test in Patients with Glaucoma and Other Ocular Conditions"

_diagnostics, 2024, doi:10.3390/diagnostics14040433_

Round 1

Reviewer 1 Report

Comments and Suggestions for Authors

The authors did not make the case why this study is needed. This test has been used for more than four decades with exceptional reliability as they also found in their study.  The amount of variation you are reporting is the nature of any test on human subjects and can not be eliminated.

The authors say this is a retrospective study. Why all tests were repeated in your setting if it was not performed for a scientific  study? Is it a routine practice in your setting to repeat the test? 

The discussion is mostly either unrelated to the subject or repetition of introduction and results section. It is unable to make the case about any significance of the study.

Comments on the Quality of English Language

Needs moderate editing.

Reviewer 2 Report

Comments and Suggestions for Authors

The authors reported the results of Binocular Esterman Visual Field Test. The design and the concept of this study were sound, and the formatting of this study was fair. Also, they point out the shortness of EVFT in which peripheral and upper test points had relatively low reliability rates. This is important in clinical practice. I think this study can be published as current form.

Reviewer 3 Report

Comments and Suggestions for Authors

Attached document

Round 2

Reviewer 1 Report

Comments and Suggestions for Authors

The article has improved. 

Comments on the Quality of English Language

Some English editing is needed.

Reviewer 3 Report

Comments and Suggestions for Authors

Thank you to the authors for their update. In my opinion, the manuscript has been substantially improved, and I think it can be published in its present form.